# TPMIL: Trainable Prototype Enhanced Multiple Instance Learning for Whole Slide Image Classification

**Litao Yang**[1]                                                    LITAO.YANG@MONASH.EDU
**Deval Mehta**[*1]                                                  DEVAL.MEHTA@MONASH.EDU
**Sidong Liu**[*2]                                                    SIDONG.LIU@MQ.EDU.AU
**Dwarikanath Mahapatra**[3]                   DWARIKANATH.MAHAPATRA@INCEPTIONIAI.ORG
**Antonio Di Ieva**[4]                                              ANTONIO.DIIEVA@MQ.EDU.AU
**Zongyuan Ge**[1]                                                 ZONGYUAN.GE@MONASH.EDU

[1] *Monash Medical AI Group, Monash University*

[2] *Australia Institute of Health Innovation, Macquarie University*

[3] *Inception Institute of Artificial Intelligence*

[4] *Computational NeuroSurgery (CNS) Lab, Macquarie Medical School, Macquarie University*

**Editors:** Accepted for publication at MIDL 2023

## Abstract

Digital pathology based on whole slide images (WSIs) plays a key role in cancer diagnosis and clinical practice. Due to the high resolution of the WSI and the unavailability of patch-level annotations, WSI classification is usually formulated as a weakly supervised problem, which relies on multiple instance learning (MIL) based on patches of a WSI. In this paper, we aim to learn an optimal patch-level feature space by integrating prototype learning with MIL. To this end, we develop a **T**rainable **P**rototype enhanced deep MIL (TPMIL) framework for weakly supervised WSI classification. In contrast to the conventional methods which rely on a certain number of selected patches for feature space refinement, we softly cluster all the instances by allocating them to their corresponding prototypes. Additionally, our method is able to reveal the correlations between different tumor subtypes through distances between corresponding trained prototypes. More importantly, TPMIL also enables to provide a more accurate interpretability based on the distance of the instances from the trained prototypes which serves as an alternative to the conventional attention score-based interpretability. We test our method on two WSI datasets and it achieves a new SOTA. GitHub repository: https://github.com/LitaoYang-Jet/TPMIL

**Keywords:** Whole Slide Image, Multiple Instance Learning, Prototype Learning

## 1. Introduction

Recent advances in digital pathology have shown the potential in disease diagnosis, medical education, and pathological research (Dimitriou et al., 2019). In particular, as the gold standard for cancer diagnosis, digital pathology can process gigapixel whole slide images (WSIs) scanned by digital slide scanners for assessment, sharing, and analysis (Li et al., 2021). Deep learning is currently in the ascendant in medical imaging and has even revolutionized this field (Esteva et al., 2019), but deep learning-based WSI analysis has always faced long-standing and unique challenges (Lu et al., 2021). The main challenge comes from extremely high resolutions of WSIs - a typical WSI generally has a size of 100K pixels at 40X

---

*\* Contributed equally*

magnification, which makes it computationally challenging to fit even in a high-end computational machine for training deep learning models (Zhang et al., 2022). Consequently, a WSI is usually first tiled into thousands of small patches (instances), then the patch-level features are extracted by deep learning models such as a CNN and finally a classifier will be used to aggregate and make the final prediction (Hou et al., 2016; Coudray et al., 2018). However, manual annotations by experienced pathologists at the patch level is a time-consuming and expensive process that is difficult to scale for big datasets and multiple pathologies. Recent works have addressed this problem by employing weakly supervised techniques based on the variants of MIL for WSI classification using only the slide-level annotations (Campanella et al., 2019; Rony et al., 2019; Li et al., 2021).

Conventionally, the standard MIL algorithms were designed for and restricted to weakly supervised positive/negative binary classification problem which assumes a WSI patches bag is labelled as positive if it contains at least one positive patch, whereas all of the patches in a negative bag should be negative (Lu et al., 2021). Such methods often consider handcrafted aggregators such as mean-pooling and max-pooling (Pinheiro and Collobert, 2015; Feng and Zhou, 2017), which are predefined and untrainable. The performance of the model will be highly dependent on the extracted features and their distribution. To address this issue, recently, some works have been proposed to enhance the patchs feature space distribution. DSMIL (Li et al., 2021) used self-supervised contrastive learning to train a better feature extractor. DGMIL (Qu et al., 2022) proposed a feature distribution modeling method that utilizes the extreme positive and negative instances and their distribution-based pseudo labels to train a binary classifier for feature space refinement. Another stream of work has proposed trainable attention-based aggregators. ABMIL (Ilse et al., 2018) proposed a trainable and interpretable attention-based pooling function that can provide an attention score to each patch and inform its contribution or importance to the bag label. CLAM (Lu et al., 2021) extended the attention-based aggregation to the general multi-class weakly supervised WSI classification and created a pipeline to generate heatmaps to further enhance the interpretability of clinical diagnoses by using attention scores. Besides, it included a technique for learning a rich and separable patch-level feature space by clustering the highest and lowest attention patches in that bag-level class.

However, only using a few extreme score patches to update the feature extractor or projector to encourage the learning of class-specific features might lead to sub-optimal patch features for weakly supervised WSI classification (Lu et al., 2019). Moreover, there is a considerable variation in image sizes and the proportion of disease-positive areas between different WSIs datasets and even within the same dataset. For instance, the proportion or number of extreme patches is usually small (e.g., 10% in DGMIL (Qu et al., 2022), eight patches in CLAM (Lu et al., 2021)). These small proportions of patches are then used to refine and separate the feature space, which can easily suffer from the overfitting or bias problem and do not optimize the feature space in an ideal way. To address these limitations, we develop a **T**rainable **P**rototype enhanced deep MIL (TPMIL) framework for weakly supervised WSI classification. Prototype learning is derived from Nearest Mean Classifiers (Guerriero et al., 2018). It aims to provide a concise representation or prototype for the entire class of instances (Sun et al., 2017; Csurka et al., 2004; Csurka and Perronnin, 2011). Some recent works on WSI such as PMIL (Yu et al., 2023), ProtoMIL (Yu et al., 2023), and (Hemati et al., 2022) have shown the potential of using representation or prototype for clas-

sification. The prototypes and image representations can be learned separately (Yu et al., 2023) or together (Yu et al., 2023) or concisely to binary and sparse (Hemati et al., 2022). These methods limit the selection of the prototypes only based on certain representative instances or patches which can lead to a wrong choice of prototypical embeddings. Moreover, these works also later use those prototypical embeddings for bag-level classification. In our work, we do not use prototypes to make the bag-level prediction but to refine the instance-level feature space, instead of training a separate instance-level classifier with extreme score patches, TPMIL integrates prototype learning with MIL by considering all the instance embeddings to learn their specific instance prototypes. By utilizing the attention-score guided prototype learning, during training, all the instances are clustered softly to their corresponding prototypes in the feature space which enables an accurate distribution of instances based on their features and not on the high-level WSI label. This enables our proposed framework to learn a more optimal patch-level feature space.

To validate our proposed approach, we evaluate TPMIL on two public WSI datasets including TCGA brain tumor dataset for multi-class classification and TCGA lung cancer dataset for binary classification. Our extensive experimental results show that TPMIL outperforms the existing MIL approaches. TPMIL makes the following contributions to the task of WSI classification - 1) Learns a more optimal patch-level feature space than existing methods which enables it to achieve a SOTA performance on two WSI datasets. 2) Provides a more accurate WSI interpretability alternative compared to the conventional approach of attention score-based WSI interpretability. 3) Enables to interpret the similarity and differences between different tumour subtypes based on the distances between the corresponding trained prototypes of the tumour subtypes.

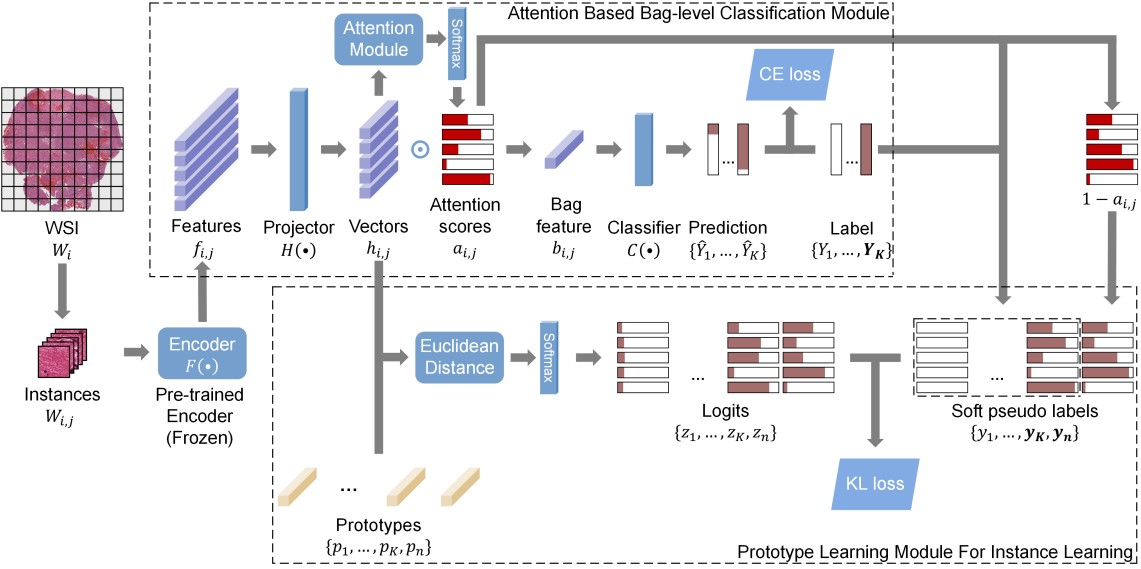

Figure 1: Overview of TPMIL. The key novelty of our proposed framework is the integration of Prototype Learning Module for learning an optimal distribution of instances.

## 2. Method

### 2.1. MIL for WSI classification

Consider a WSI dataset $W = \{W_i\}_{i=1}^N$ which consists of $N$ WSIs. In the MIL approach, each WSI $W_i$ is considered as a bag which contains $M$ small patches $W_i = \{W_{i,j}\}_{j=1}^M$ tiled from $W_i$, where $M$ can vary for each $W_i$ and each patch is also referred to as an instance of the $W_i$. In the case of multi-class subtyping problem with $K$ classes, where a $W_i$ is annotated as one of the categories in $Y \in \{Y_1, Y_2, \ldots, Y_K\}$, it is assumed that the bag consists of type $Y$ tumor patches and normal tissue patches. However, there is no access to instance labels and there are only bag-level labels $Y$ available in the weakly supervised WSI classification task. To predict the labels of WSIs, MIL further uses an encoder $F(\cdot)$ to extract patch features and a transformation $G(\cdot)$ to aggregate the bag feature. Finally, a bag classifier $C(\cdot)$ will be used to predict the WSI $\widehat{Y}$, given by Equation (1).

$$\widehat{Y} = C\left(G\left(F\left(W_{i,1}\right), F\left(W_{i,2}\right), \ldots, F\left(W_{i,M}\right)\right)\right) \tag{1}$$

### 2.2. Framework Overview

Figure 1 shows the overall framework of TPMIL. First, we use a pre-trained feature extractor $F(\cdot)$ to compute the feature representations $f_{i,j}$ for all the patches $W_{i,j}$ tiled from each slide. Then the attention based bag-level classification module will uitilize the instances features to get the attention scores and predict the WSI label where the cross-entropy (CE) loss is used for model training. Meanwhile, the attention scores and WSI label are combined as soft pseudo labels which are passed into the prototype learning module for feature space refinement by employing Kullback–Leibler divergence (KLD) loss. A detailed description of the attention based bag-level classification module and the prototype learning module are presented in section 2.3 and 2.4.

### 2.3. Attention Based Bag-level Classification Module

In this module, we use the attention-based aggregation method (Ilse et al., 2018; Lu et al., 2021) for bag prediction. After getting the instances features $f_{i,j}$, a projector $H(\cdot)$ (a fully connected layer) is used to project $f_{i,j}$ into vectors $h_{i,j}$, which are then passed to an attention module to generates the attention score $a_{i,j}$ for each instance $W_{i,j}$. The gated attention mechanism from ABMIL (Ilse et al., 2018) is employed in the attention module. It consists of three fully connected layers - $U, V$ and $w$, which combine gating mechanism (Dauphin et al., 2017) and the hyperbolic tangent $\tanh(\cdot)$ element-wise non-linearity, given by Equation (2),

$$a_{i,j} = \frac{\exp\left\{w^\top\left(\tanh\left(Vh_{i,j}^\top\right) \odot sigm\left(Uh_{i,j}^\top\right)\right)\right\}}{\sum_{m=1}^M \exp\left\{w^\top\left(\tanh\left(Vh_{i,m}^\top\right) \odot sigm\left(Uh_{i,m}^\top\right)\right)\right\}} \tag{2}$$

where $\odot$ is an element-wise multiplication and $sigm(\cdot)$ is the sigmoid non-linearity. The normalized (by Softmax) attention scores $a_{i,j}$ multiplied by the vectors $h_{i,j}$ are then aggregated into the bag feature $b_{i,j}$ and passed to the classifier $C(\cdot)$ to make the prediction $\left\{\widehat{Y}_1, \ldots, \widehat{Y}_K\right\}$ for the WSI slide.

## 2.4. Prototype Learning Module

To enhance and optimize the feature space by considering all the instances and their feature distributions instead of extreme samples that may lead to bias, we propose and employ Prototype Learning Module which could provide a better learning strategy for the instances. For tumor WSIs, we consider that all classes are mutually exclusive and only two types of instances will be included in a single WSI - either the tumor subtype (positive) instance or the normal tissue (negative) instance and all negative instances share a similar feature distribution. So there will be $K + 1$ (K tumor subtypes and one normal tissue) classes of instances in WSI dataset containing $K$ subtypes of tumor. Thus, in TPMIL, we train $K+1$ prototypes $\{p_1, p_2, \ldots, p_K, p_n\}$ which include $K$ tumor subtypes instance prototypes and one negative instance (normal tissue) prototype. Except for these $K + 1$ trainable prototypes, there are no other trainable parameters in the prototype learning module. The prototypes have the same vector size as a single instance vector $h_{i,j}$ and will be randomly initialized before training. We first calculate the euclidean distance between each instance and each prototype separately. Each instance will have $K + 1$ distances which indicate its similarity to those prototypes (the smaller the distance, the more similar). We then convert these distances to $K + 1$ logits $\{z_1, \ldots, z_K, z_n\}$ by applying the softmax layer. As we don't have the instance labels, instead of labelling each instance corresponding to the WSI label, we utilize the attention score $a_{i,j}$ of the instance to generate its soft pseudo label by multiplying the normalized attention score (ranging from 0 to 1) of the instance with the WSI label. For example, in Figure 1, considering a WSI image having a label $Y_K$, the WSI image will only have two instances - either $Y_K$ or negative (normal). Thus, the attention $(a_{i,j})$ score of the instance is multiplied with the WSI label, whereas the leftover of the attention score $(1 - a_{i,j})$ is multiplied with the normal tissue category to generate its soft pseudo labels $\{y_1, \ldots, y_K, y_n\}$. We employ the Kullback–Leibler divergence loss between the soft pseudo label and the logits of the instances for training the prototype learning module. This way we develop the strategy to consider all the instances and their weightage in the training by employing attention-guided prototype learning.

As the prototype learning module is trained together with the whole model, the instance feature space is refined optimally by the prototype learning. During the training, the prototypes will be updated to find a concise representation of the different categories. With our employed soft pseudo label strategy, for a WSI image with tumor subtype $Y_K$, the instances with high attention scores will be forced to be closer to the corresponding class prototype $p_K$, while those with the low attention scores instances will be forced to be closer to the negative prototype $p_n$. Our strategy will also ensure that all the instances in that bag of $Y_K$ will be far away from prototypes of the remaining categories. Thus, our proposed framework will softly cluster all the instances and optimally refine the feature space.

## 3. Experiment and Results

### 3.1. Dataset

#### 3.1.1. TCGA Brain Tumor dataset

A brain tumor WSI dataset was acquired from The Cancer Genome Atlas (TCGA) (Clark et al., 2013), which is a publicly available repository of H&E stained WSIs and multi-omics

data. This brain tumor WSI dataset includes three subtypes of gliomas, namely Astrocytoma, Oligodendroglioma and Glioblastoma. The Formalin-Fixed Paraffin-Embedded (FFPE) slides were acquired for the patients for analysis, as FFPE slides are the current gold standard for brain tumor diagnostics and more suitable for computational analysis compared to frozen slides (Jose et al., 2022). The number of samples amongst different categories is distributed as - 183 Astrocytoma slides, 335 Oligodendroglioma slides and 630 Oligodendroglioma slides, with slide-level labels available.

### 3.1.2. TCGA Lung Cancer dataset

The TCGA Lung Cancer dataset includes two sub-types of lung cancer, namely Lung Adenocarcinoma (LUAD) and Lung Squamous Cell Carcinoma (LUSC). The dataset contains 534 LUAD slides and 512 LUSC slides with only slide-level labels available, which is a public dataset and can be downloaded from National Cancer Institute Data Portal.

### 3.2. Implementation Details

For the experiments on the TCGA Brain Tumor dataset, we use a ResNet50 (He et al., 2016) deep learning model pre-trained on ImageNet (Russakovsky et al., 2015) as the feature extractor and convert each patch into a 1024-dimensional vector. We perform five-fold cross-validation to evaluate our framework on this dataset and report the average performance of the test set for the five experiments. Since the patient IDs are available, our data splits are based on the patient level to avoid bias. Specifically, we first randomly split the entire dataset into five folds. In each experiment, we select one fold as the test set and the remaining data will be split into 80% training set and 20% validation set. We use the Adam optimizer with a fixed learning rate of 0.0002 and weight decay of 0.00001 to optimize the model parameters with 200 epochs. Meanwhile, we employ the weighted sampling strategy during training for this dataset. For the experiments on the TCGA Lung Cancer dataset, we use the pre-trained feature extractor provided by DSMIL (Li et al., 2021) and convert each patch into a 512-dimensional vector. For the TCGA Lung Cancer dataset, we use the same settings of data split as DSMIL (Li et al., 2021) and DGMIL (Qu et al., 2022) to make a fair comparison. The other experimental settings of learning rate, weight decay, and number of epochs are the same as above. We don't employ any other tricks for performance improvement and all our experiments are performed on an NVIDIA RTX A5000.

### 3.3. Quantitative Results

Table 1 shows the benchmarking of different methods on the TCGA Brain Tumor dataset and TCGA Lung Cancer dataset. We employ the test metrics of AUC and ACC for comparison purposes. To fairly compare the performance improvement achieved by our prototype learning module, we also implement TPMIL without it and it is termed as the baseline method in Table 1. From Table 1, we can note that CLAM (Lu et al., 2021) shows marginal performance improvements compared with baseline - 0.01% higher AUC and 0.26% higher ACC in TCGA Brain Tumor dataset and the similar trend in TCGA Lung Cancer dataset. This performance improvement indicates the limited effects achieved on the feature space refinement by only clustering the extreme score instances. In contrast, TPMIL considers all the instance embeddings by prototype learning to enhance the feature space. Compared

Table 1: Comparison of performances on both datasets

| Dataset | Method | AUC | ACC |
|---|---|---|---|
| | Baseline | 0.9393 | 0.8162 |
| TCGA Brain Tumor dataset | CLAM | 0.9394 | 0.8188 |
| | TPMIL(Ours) | **0.9417** | **0.8316** |
| | Baseline | 0.9783 | 0.9381 |
| | DSMIL | 0.9633 | 0.9190 |
| TCGA Lung Cancer dataset | DGMIL | 0.9702 | 0.9200 |
| | CLAM | 0.9788 | 0.9286 |
| | TPMIL(Ours) | **0.9799** | **0.9427** |

with CLAM, our method achieves **0.24%** higher AUC and **1.54%** higher ACC in the TCGA Brain Tumor dataset and **0.11%** higher AUC and **1.41%** higher ACC as well as **SOTA** performance in TCGA Lung Cancer dataset. This proves to a large extent the superiority of our approach and its specific advantages of refining the feature space.

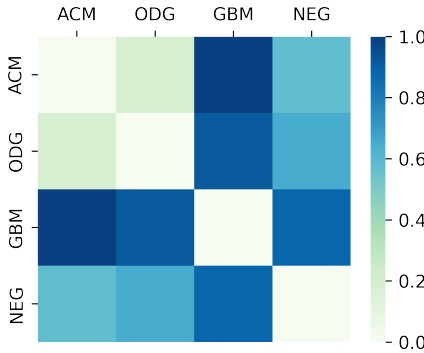

Figure 2: The distance matrix of TCGA Brain Tumor prototypes after training

### 3.4. Visualization Results

#### 3.4.1. Inter Prototype-Distance matrix

Figure 2 shows the distances between different prototype centres (ACM = Astrocytoma; ODG = Oligodendroglioma; GBM = Glioblastoma; NEG = Negative) where darker colour indicates higher distance and lower correlation. The distance between ACM and ODG is markedly lower than their distance to GBM, as both ACM and ODG are lower grade gliomas (LGG) whereas GBM is higher grade glioma (HGG), which may have distinct morphological features, such as necrosis and pseudo palisading cells around necrosis. The NEG prototype aims to capture the regions that do not contribute to the classification such as normal tissues, blood vessels and artefacts. As shown in Figure 2, NEG is closer to ACM and ODG compared to GBM. This might be attributable to the fact that GBM has more

diverse morphological features, and that the NEG prototype accounts for a much smaller proportion of tissues in the WSI.

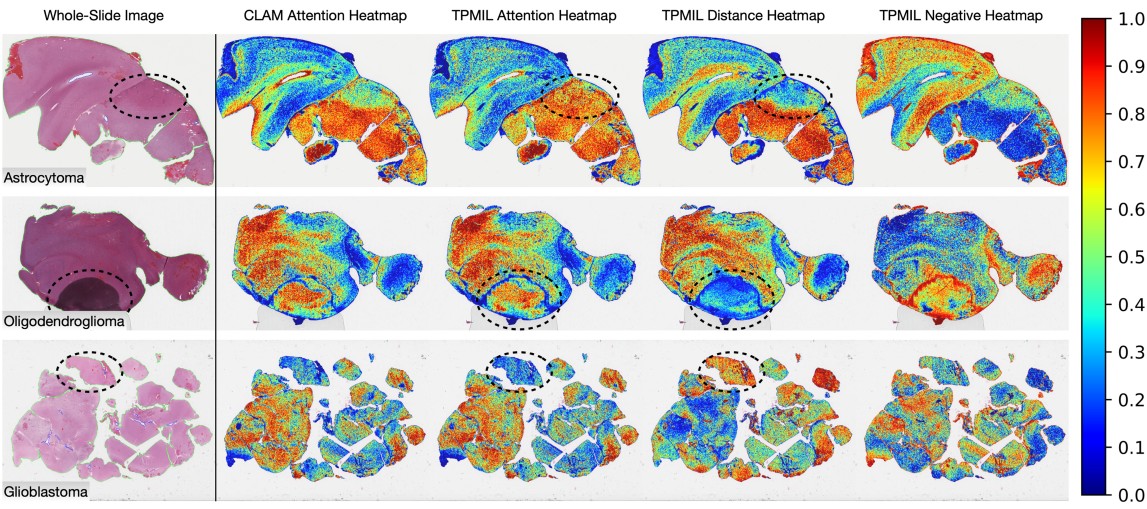

Figure 3: Interpretability and visualization of heatmaps for brain tumor WSI classification.

### 3.4.2. Interpretability of WSI Images

Since the ground truth is not unavailable, we consult expert clinicians who helped to evaluate and compare the interpretability of the different heatmaps shown in Figure 3. A representative slide from each subtype is shown in Figure 3 ($1^{st}$ column), with its corresponding whole-slide heatmaps generated by computing attention scores for the predicted class of the model over patches ($2^{nd}$ column: CLAM Attention Heatmap; $3^{rd}$ column: the proposed TPMIL Attention Heatmap), and by computing the distances of the patches to the prototype centres using TPMIL ($4^{th}$ column: distance to the predicted class prototype centre; $5^{th}$ column: distance to the negative class prototype centre). The CLAM and the proposed TPMIL method give similar attention heatmaps which could highlight the tumour-normal tissue boundary. Patches of the most highly attended regions (in Attention Heatmaps) and the regions in close proximity of the prototype centres (in Distance Heatmaps) generally exhibit well-known tumour morphology, e.g., high tumour cellularity and necrosis.

The differences between attention heatmaps and distance heatmaps are highlighted in the circled region in Figure 3. In Astrocytoma and Oligodendroglioma, the circled regions are **artefacts** which should not be considered as positively related to the diagnosis, and it can be seen that Distance heatmap is more robust to **artefacts** as it doesn't give a dark red colour. In Glioblastoma, the circled region is a **necrotic** area which is a diagnostic feature of glioblastoma and Distance Heatmap successfully captured it by indicating the dark red colour in that region. In contrast to the distance heatmap from the predicted tumour type, the negative prototype distance heatmaps ($5^{th}$ column) highlight the regions that are close to the negative prototype centres, including normal brain tissues and blood vessels, among

different background artefacts such as the overstained regions in the Oligodendroglioma WSI. This proves the effectiveness of the heatmaps based on the prototype distance which can serve as an important alternative to the conventional attentional heatmaps employed for explainability purposes. The enhanced interpretability with prototype distance heatmap can really help clinicians to provide a reliable reference for analysing.

## 4. Conclusion

In this paper, we present TPMIL: a Trainable Prototype enhanced MIL framework with a novel feature space refinement strategy for weakly supervised WSI classification. Compared with existing MIL methods, our method is able to provide a better feature space representation by considering all the instances rather than some selected ones. Our results on two WSIs datasets demonstrate the superior performance and validity of our method which achieves a new SOTA. Our method further improves interpretability by employing distance-based visualization heatmaps which acts as an alternative to the conventional attention score-based interpretability. We believe our work will provide a solid step forward for the research community in MIL-based weakly supervised WSI classification task.

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
