# OpenReview forum: "TPMIL: Trainable Prototype Enhanced Multiple Instance Learning for Whole Slide Image Classification"
_MIDL.io/2023/Conference — MIDL 2023 Poster_

### Official Review · Reviewer_3Gx5 · 2023-02-01

**Confidence:** 4
**Preliminary Rating:** 4
**Recommendation:** Poster

**Summary:**

The authors have proposed a method named - Trainable Prototype enhanced deep MIL (TPMIL) method for weakly supervised WSI classification. Along with the popular attention-based aggregation method for MIL problems, they have incorporated a trainable prototype module for better learning the patch-level feature representation. The authors have evaluated their approach on two different datasets - TCGA Brain Tumor dataset for multi-class classification and the TCGA Lung Cancer dataset for binary classification. Further, the authors have used the distance between disease sub-type prototype representations for interpretability and have shared the distance matrix and heatmaps to highlight their findings.

**Strengths:**

- Overall, the paper is well-structured. The authors have clearly shared their motivations and have cited relevant papers.
- The prototype learning module is an interesting contribution, enabling the learning of optimal sub-type feature representation. Empirically, authors have demonstrated that their approach is superior to focusing only on extreme patches. Furthermore, they have used soft pseudo-labels to learn signals from all the instances, hence appropriately weighting patches based on their attention score.
- The findings shared in the visualization section are relevant. Inter prototype-distance matrix aids in a better understanding of disease sub-types and their relative similarities. Also, heatmaps based on distances are robust to artifacts.


**Weaknesses:**

- In the 2.3 section, it is not clear how the attention score is normalized. Is it by dividing with maximum attention score, or any other strategy is used?
- It is unclear in the diagram if the soft-pseudo-label attention score represents the instance label or the WSI label. It might be helpful if, in the prototype learning module part, authors highlight the learning scenario for a specific instance.
- It is unclear which part of the encoder is trained using the prototype learning module. As per my understanding, it is not computationally feasible to train the encoder and projector, both with prototype learning modules for all the WSI patches. It will be helpful if authors can specify what part is frozen and not updated during training.
- It is slightly misleading only to use the DSMIL feature extractor for the Lung Cancer dataset comparison because the feature extractor plays a major role in performance improvement. And currently, the contribution of domain-based feature encoders and prototype modules needs to be clarified. The authors should add a comparison with the Imagenet encoder and should specify if, for CLAM, the DSMIL feature encoder is used or Imagenet. Further, authors should highlight if there are any ways of fine-tuning the encoder along with the projector since a part of performance gain also comes from learning domain-specific encoder.


**Deanonymize Review:**

no

**Detailed Comments:**

Kindly let me know if I have misunderstood or missed any part of the paper. It will be helpful also to include those details in the paper as it will improve the readability and understanding of the approach. Hopefully, authors will open-source the code in the future.

**Paper Type:**

methodological development

**Questions To Address In The Rebuttal:**

Kindly address the concerns highlighted in the weakness section, and provide more details about the approach in the paper. It will be helpful if the authors can update the diagram, provide more details about attention score normalization, specify the frozen and unfrozen part of the architecture, and clarify the contribution of the DSMIL feature encoder and prototype module.

---

### Official Review · Reviewer_Rb6Y · 2023-02-02

**Confidence:** 3
**Preliminary Rating:** 4
**Recommendation:** Poster

**Summary:**

This paper targets the WSI representation learning task. Authors propose to use Trainable Prototypes to improve the current enhanced deep MIL methods for weakly supervised WSI classification. The method is able to have more accurate interpretability as it can capture the correlations between different tumour subtypes.



**Strengths:**

1. The idea is clear.
2. The presentation is good and the paper is well-written.
3. The method offers more reliable interpretability.
4. Employing prototype learning has improved the performance of attention based MIL


**Weaknesses:**

1. The technical novelty is limited. The paper basically proposes to combine existing ideas for set representation learning.
2. The experiment section of the paper is weak. It would be nice if the authors evaluate the idea on images from another dataset (NOT TCGA)
3. The proposed method seems to be non-end-to-end.

**Deanonymize Review:**

no

**Detailed Comments:**

The idea of prototype learning for set representation learning is very similar to the Bag of Visual Words and its extensions like the Fisher vector technique used in old-school computer vision to combine a set of local descriptors into one global embedding per image. However, the literature review in this part doesn't exist. I would like to see more discussion on the related works of prototype learning and their connection to this work. Especially the main papers that have not been discussed:

1. Csurka, G., Dance, C., Fan, L., Willamowski, J., Bray, C.: Visual categorization with bags of
keypoints. In: ECCV Workshop on Statistical Learning for Computer Vision (2004)

2. Csurka, Gabriela, and Florent Perronnin. "Fisher vectors: Beyond bag-of-visual-words image representations." Computer Vision, Imaging and Computer Graphics. Theory and Applications: International Joint Conference, VISIGRAPP 2010, Angers, France, May 17-21, 2010. Revised Selected Papers. Springer Berlin Heidelberg, 2011.

Further, the deep learning-based Fisher Vector also has been proposed where parameters of the deep generative model play the role of the prototype. This technique has been also employed in the following paper for the WSI representation learning task:

3. Hemati, Sobhan, et al. "Learning Binary and Sparse Permutation-Invariant Representations for Fast and Memory Efficient Whole Slide Image Search." arXiv preprint arXiv:2208.13653 (2022).

It would be good if the authors discuss the connection of the above works to their work.



**Paper Type:**

methodological development

**Questions To Address In The Rebuttal:**

1. I wonder how many patches per WSI have been used for training and how this number may affect the performance.
2. I there a way to modify the method to be end-to-end? what are the bottlenecks to doing so?

---

### Official Review · Reviewer_4k7B · 2023-02-03

**Confidence:** 4
**Preliminary Rating:** 3

**Summary:**

This paper presents a classification method for WSI. It utilizes the principle of prototype and tries to learn such prototypes for each class.
The KL divergence loss is applied between the network output and distance (to prototypes) after softmax.
The experiments show the best performance compared with others.


**Strengths:**

- This paper is well-written and easy to follow. It has a clear structure.
- The method sounds make sense.
- The performance outperforms other attention methods.
- The visualization map with heatmap looks really good.


**Weaknesses:**

- There are also some other papers utilizing prototype information for the WSI classification, like [1]. This paper misses a background description of the prototype-based WSI classification in the Introduction part. And also missing the difference from previous prototype-based methods.
- The improvement of the performance is marginal.
- Fig.3 misses the colour bar to indicate the value of different colours. Is it possible here to show the ground truth for comparison?



[1]. Yu, Jin-Gang, et al. "Prototypical multiple instance learning for predicting lymph node metastasis of breast cancer from whole-slide pathological images." Medical Image Analysis (2023): 102748.

**Deanonymize Review:**

no

**Paper Type:**

methodological development

**Questions To Address In The Rebuttal:**

- There are also some other papers utilizing prototype information for the WSI classification, like [1]. This paper misses a background description of the prototype-based WSI classification in the Introduction part. And also missing the difference from previous prototype-based methods.
- The improvement of the performance is marginal.
- Fig.3 misses the colour bar to indicate the value of different colours. Is it possible here to show the ground truth for comparison?



[1]. Yu, Jin-Gang, et al. "Prototypical multiple instance learning for predicting lymph node metastasis of breast cancer from whole-slide pathological images." Medical Image Analysis (2023): 102748.

---

### Official Review · Reviewer_5XRM · 2023-02-06

**Confidence:** 3
**Preliminary Rating:** 3

**Summary:**

This work extended multiple instance learning (MIL) by introducing prototype learning to learn a more optimal patch-level feature space. The learned prototypes can also be used to interpret the similarity and differences between different tumour subtypes. Experiments have been done on two whole slide images (WSIs) datasets and showed marginal improvements compared with previous states of the art.

**Strengths:**

1. The idea to integrate MIL and prototype learning is intuitive and intriguing.
2. Using prototypes to interpret the similarity and differences between different tumour subtypes (Fig. 2) is interesting.

**Weaknesses:**

1. The method, especially the prototype learning module, is not well explained. It is not clear what kind of role the prototype learning module play during the inference stage.
2. The experimental setting is not clear (see "Questions To Address In The Rebuttal" for details).
3. The improvement is only marginal compared with existing SOTAs.

**Deanonymize Review:**

no

**Detailed Comments:**

Please refer to the "Questions To Address In The Rebuttal" section.

**Paper Type:**

methodological development

**Questions To Address In The Rebuttal:**

1. Method
1.a It is not clear how prototypes are used to make the bag-level classification (i.e., \hat{Y}). Are the Vectors (h_{i,j}) calculated based on the prototypes?
1.b Figure 1 is not very illustrative. Particularly, what do the one-side arrow and the double-side arrow (e.g., the arrow between Vectors and Prototypes) mean?

2. Experiments
2.a There are two modules in the proposed TPMIL: attention-based aggregation module and prototype learning module. According to Section 2.3, the attention-based aggregation module is based on CLAM (Lu et al., 2021). Therefore, what is the difference between the Baseline (w/o prototype learning module) and CLAM?
2.b In Figure 3, it seems TPMIL Attention Heatmaps and Distance Heatmaps may highlight different parts of the same input. Therefore, in the testing phase, how could a user decide the correct regions to focus on giving the two different heatmaps from TPMIL.
2.c In Figure 3, please make it clear whether the circled region is positively related to the ground truth on each heatmap.

3. Writing
3.a In Page 3, the author claimed that "To the best of our knowledge, we are the first to integrate prototype learning with MIL for feature space refinement." However, several exciting works have explored this direction, including [1] and [2]. It would be better to change the claim and briefly discuss the difference between the proposed method and [1,2].

[1] Rymarczyk, D., Pardyl, A., Kraus, J., Kaczyńska, A., Skomorowski, M. and Zieliński, B., 2021. ProtoMIL: Multiple Instance Learning with Prototypical Parts for Whole-Slide Image Classification. arXiv preprint arXiv:2108.10612.
[2] Yu, J.G., Wu, Z., Ming, Y., Deng, S., Li, Y., Ou, C., He, C., Wang, B., Zhang, P. and Wang, Y., 2023. Prototypical multiple instance learning for predicting lymph node metastasis of breast cancer from whole-slide pathological images. Medical Image Analysis, p.102748.

The revised manuscript showed a lot of improvements after the rebuttal, especially in the discussion of interpretability on Page 8. Therefore, I would like to change my rating from Weak Reject to Borderline.

---

### Meta-Review · Area_Chair_zZym · 2023-02-23

**Recommendation:** Accept (Poster)
**Confidence:** 4

**Metareview:**

This paper proposed to learn an optimal patch-level feature space by integrating prototype learning with MIL. Experimental results on two WSI datasets demonstrated improved performance.
Here I summarize the key contributions and limitations.

Pros:
The combination of MIL and prototype learning is interesting;
 Improved interpretability along with classification accuracy;
The paper is well-written.

Cons:
The methodological details are not well explained;
Unclear experimental settings and marginal improvements.

In summary, most reviewers confirmed the merits of introducing prototype learning with MIL for improving the interpretation and recognition performance. Other concerns mentioned by reviewers can be addressed properly in the final version. Therefore, I recommend the decision of accept.